# FRUSTRATINGLY SHORT ATTENTION SPANS IN NEURAL LANGUAGE MODELING

**Michał Daniluk, Tim Rocktäschel, Johannes Welbl & Sebastian Riedel**
Department of Computer Science
University College London
`michal.daniluk.15@ucl.ac.uk`,
`{t.rocktaschel,j.welbl,s.riedel}@cs.ucl.ac.uk`

## ABSTRACT

Neural language models predict the next token using a latent representation of the immediate token history. Recently, various methods for augmenting neural language models with an attention mechanism over a differentiable memory have been proposed. For predicting the next token, these models query information from a memory of the recent history which can facilitate learning mid- and long-range dependencies. However, conventional attention mechanisms used in memory-augmented neural language models produce a single output vector per time step. This vector is used both for predicting the next token *as well as* for the key and value of a differentiable memory of a token history. In this paper, we propose a neural language model with a key-value attention mechanism that outputs separate representations for the key and value of a differentiable memory, as well as for encoding the next-word distribution. This model outperforms existing memory-augmented neural language models on two corpora. Yet, we found that our method mainly utilizes a memory of the five most recent output representations. This led to the unexpected main finding that a much simpler model based only on the concatenation of recent output representations from previous time steps is on par with more sophisticated memory-augmented neural language models.

## 1 INTRODUCTION

At the core of language models (LMs) is their ability to infer the next word given a context. This requires representing context-specific dependencies in a sequence across different time scales. On the one hand, classical $N$-gram language models capture relevant dependencies between words in short time distances explicitly, but suffer from data sparsity. Neural language models, on the other hand, maintain and update a dense vector representation over a sequence where time dependencies are captured implicitly (Mikolov et al., 2010). A recent extension of neural sequence models are attention mechanisms (Bahdanau et al., 2015), which can capture long-range connections more directly. However, we argue that applying such an attention mechanism directly to neural language models requires output vectors to fulfill several purposes at the same time: they need to (i) encode a distribution for predicting the next token, (ii) serve as a key to compute the attention vector, as well as (iii) encode relevant content to inform future predictions.

We hypothesize that such overloaded use of output representations makes training the model difficult and propose a modification to the attention mechanism which separates these functions explicitly, inspired by Miller et al. (2016); Ba et al. (2016); Reed & de Freitas (2015); Gulcehre et al. (2016). Specifically, at every time step our neural language model outputs three vectors. The first is used to encode the next-word distribution, the second serves as key, and the third as value for an attention mechanism. We term the model *key-value-predict* attention and show that it outperforms existing memory-augmented neural language models on the Children's Book Test (CBT, Hill et al., 2016) and a new corpus of 7500 Wikipedia articles. However, we observed that this model pays attention mainly to the previous five memories. We thus also experimented with a much simpler model that only uses a concatenation of output vectors from the previous time steps for predicting the next token. This simple model is on par with more sophisticated memory-augmented neural language models. Thus, our main finding is that modeling short attention spans properly works well and provides notable

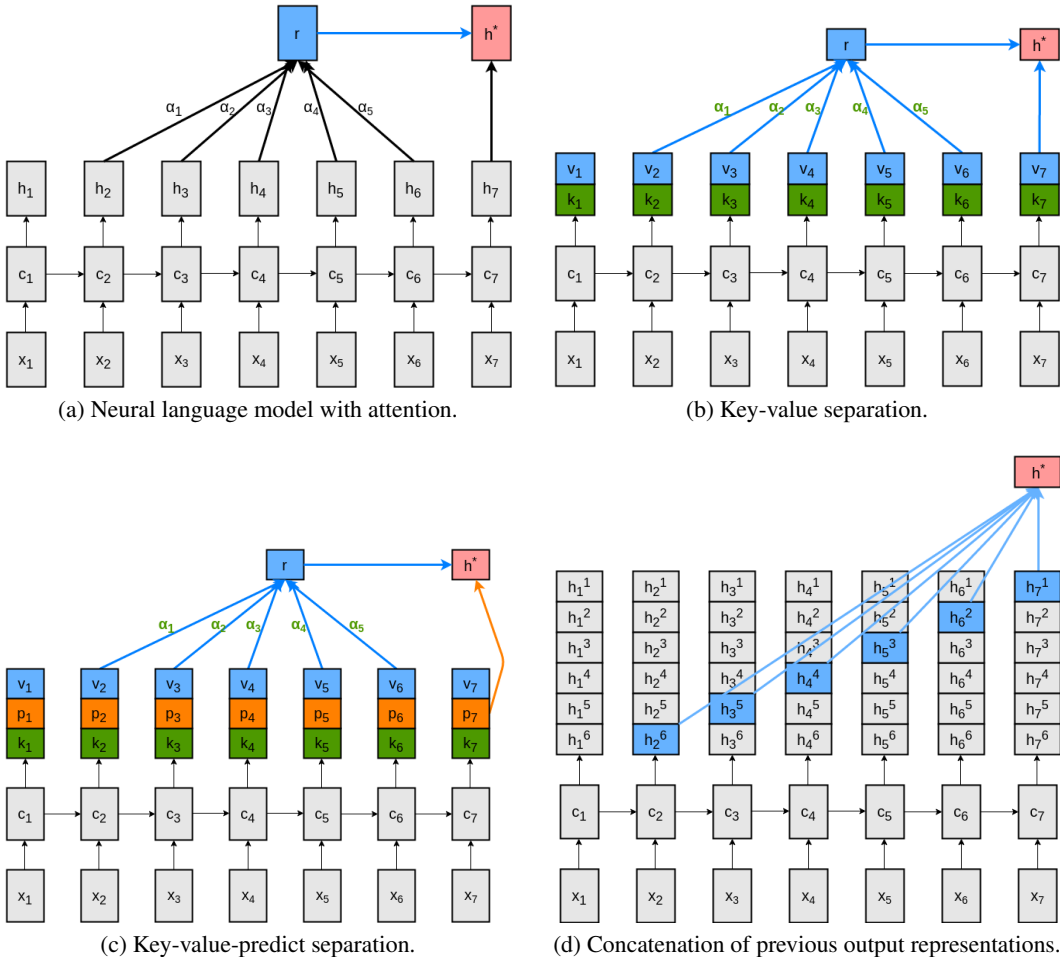

(a) Neural language model with attention.

(b) Key-value separation.

(c) Key-value-predict separation.

(d) Concatenation of previous output representations.

Figure 1: Memory-augmented neural language modelling architectures.

improvements over a neural language model with attention. Conversely, it seems to be notoriously hard to train neural language models to leverage long-range dependencies.

In this paper, we investigate various memory-augmented neural language models and compare them against previous architectures. Our contributions are threefold: (i) we propose a key-value attention mechanism that uses specific output representations for querying a sliding-window memory of previous token representations, (ii) we demonstrate that while this new architecture outperforms previous memory-augmented neural language models, it mainly utilizes a memory of the previous five representations, and finally (iii) based on this observation we experiment with a much simpler but effective model that uses the concatenation of three previous output representations to predict the next word.

## 2 METHODS

In the following, we discuss methods for extending neural language models with differentiable memory. We first present a standard attention mechanism for language modeling (§2.1). Subsequently, we introduce two methods for separating the usage of output vectors in the attention mechanism: (i) using a dedicated key and value (§2.2), and (ii) further separating the value into a memory value and a representation that encodes the next-word distribution (§2.3). Finally, we describe a very simple method that concatenates previous output representations for predicting the next token (§2.4).

## 2.1 ATTENTION FOR NEURAL LANGUAGE MODELING

Augmenting a neural language model with attention (Bahdanau et al., 2015) is straight-forward. We simply take the previous $L$ output vectors as memory $\boldsymbol{Y}_t = [\boldsymbol{h}_{t-L} \cdots \boldsymbol{h}_{t-1}] \in \mathbb{R}^{k \times L}$ where $k$ is the output dimension of a Long Short-Term Memory (LSTM) unit (Hochreiter & Schmidhuber, 1997). This memory could in principle contain all previous output representations, but for practical reasons we only keep a sliding window of the previous $L$ outputs. Let $\boldsymbol{h}_t \in \mathbb{R}^k$ be the output representation at time step $t$ and $\mathbf{1} \in \mathbb{R}^L$ be a vector of ones.

The attention weights $\boldsymbol{\alpha} \in \mathbb{R}^L$ are computed from a comparison of the current and previous LSTM outputs. Subsequently, the context vector $\boldsymbol{r}_t \in \mathbb{R}^k$ is calculated from a sum over previous output vectors weighted by their respective attention value. This can be formulated as

$$\boldsymbol{M}_t = \tanh(\boldsymbol{W}^Y \boldsymbol{Y}_t + (\boldsymbol{W}^h \boldsymbol{h}_t)\mathbf{1}^T) \qquad \in \mathbb{R}^{k \times L} \qquad (1)$$

$$\boldsymbol{\alpha}_t = \mathrm{softmax}(\boldsymbol{w}^T \boldsymbol{M}_t) \qquad \in \mathbb{R}^{1 \times L} \qquad (2)$$

$$\boldsymbol{r}_t = \boldsymbol{Y}_t \boldsymbol{\alpha}^T \qquad \in \mathbb{R}^k \qquad (3)$$

where $\boldsymbol{W}^Y, \boldsymbol{W}^h \in \mathbb{R}^{k \times k}$ are trainable projection matrices and $\boldsymbol{w} \in \mathbb{R}^k$ is a trainable vector. The final representation that encodes the next-word distribution is computed from a non-linear combination of the attention-weighted representation $\boldsymbol{r}_t$ of previous outputs and the final output vector $\boldsymbol{h}_t$ via

$$\boldsymbol{h}_t^* = \tanh(\boldsymbol{W}^r \boldsymbol{r}_t + \boldsymbol{W}^x \boldsymbol{h}_t) \qquad \in \mathbb{R}^k \qquad (4)$$

where $\boldsymbol{W}^r, \boldsymbol{W}^x \in \mathbb{R}^{k \times k}$ are trainable projection matrices. An overview of this architecture is depicted in Figure 1a. Lastly, the probablity distribution $\boldsymbol{y}_t$ for the next word is represented by

$$\boldsymbol{y}_t = \mathrm{softmax}(\boldsymbol{W}^* \boldsymbol{h}_t^* + \boldsymbol{b}) \qquad \in \mathbb{R}^{|V|} \qquad (5)$$

where $\boldsymbol{W}^* \in \mathbb{R}^{|V| \times k}$ and $\boldsymbol{b} \in \mathbb{R}^{|V|}$ are a trainable projection matrix and bias, respectively.

## 2.2 KEY-VALUE ATTENTION

Inspired by Miller et al. (2016); Ba et al. (2016); Reed & de Freitas (2015); Gulcehre et al. (2016), we introduce a key-value attention model that separates output vectors into keys used for calculating the attention distribution $\boldsymbol{\alpha}_t$, and a value part used for encoding the next-word distribution and context representation. This model is depicted in Figure 1b. Formally, we rewrite Equations 1-4 as follows:

$$\begin{bmatrix} \boldsymbol{k}_t \\ \boldsymbol{v}_t \end{bmatrix} = \boldsymbol{h}_t \qquad \in \mathbb{R}^{2k} \qquad (6)$$

$$\boldsymbol{M}_t = \tanh(\boldsymbol{W}^Y [\boldsymbol{k}_{t-L} \cdots \boldsymbol{k}_{t-1}] + (\boldsymbol{W}^h \boldsymbol{k}_t)\mathbf{1}^T) \qquad \in \mathbb{R}^{k \times L} \qquad (7)$$

$$\boldsymbol{\alpha}_t = \mathrm{softmax}(\boldsymbol{w}^T \boldsymbol{M}_t) \qquad \in \mathbb{R}^{1 \times L} \qquad (8)$$

$$\boldsymbol{r}_t = [\boldsymbol{v}_{t-L} \cdots \boldsymbol{v}_{t-1}] \boldsymbol{\alpha}^T \qquad \in \mathbb{R}^k \qquad (9)$$

$$\boldsymbol{h}_t^* = \tanh(\boldsymbol{W}^r \boldsymbol{r}_t + \boldsymbol{W}^x \boldsymbol{v}_t) \qquad \in \mathbb{R}^k \qquad (10)$$

In essence, Equation 7 compares the key at time step $t$ with the previous $L$ keys to calculate the attention distribution $\boldsymbol{\alpha}_t$ which is then used in Equation 9 to obtain a weighted context representation from values associated with these keys.

## 2.3 KEY-VALUE-PREDICT ATTENTION

Even with a key-value separation, a potential problem is that the same representation $\boldsymbol{v}_t$ is still used both for encoding the probability distribution of the next word and for retrieval from the memory via the attention later. Thus, we experimented with another extension of this model where we further separate $\boldsymbol{h}_t$ into a *key*, a *value* and a *predict* representation where the latter is only used for encoding the next-word distribution (see Figure 1c). To this end, equations 6 and 10 are replaced by

$$\begin{bmatrix} \boldsymbol{k}_t \\ \boldsymbol{v}_t \\ \boldsymbol{p}_t \end{bmatrix} = \boldsymbol{h}_t \qquad \in \mathbb{R}^{3k} \qquad (11)$$

$$\boldsymbol{h}_t^* = \tanh(\boldsymbol{W}^r \boldsymbol{r}_t + \boldsymbol{W}^x \boldsymbol{p}_t) \qquad \in \mathbb{R}^k \qquad (12)$$

More precisely, the output vector $\boldsymbol{h}_t$ is divided into three equal parts: *key*, *value* and *predict*. In our implementation we simply split the output vector $\boldsymbol{h}_t$ into $\boldsymbol{k}_t$, $\boldsymbol{v}_t$ and $\boldsymbol{p}_t$. To this end the hidden dimension of the key-value-predict attention model needs to be a multiplicative of three. Consequently, the dimensions of $\boldsymbol{k}_t$, $\boldsymbol{v}_t$ and $\boldsymbol{p}_t$ are 100 for a hidden dimension of 300.

## 2.4 $N$-GRAM RECURRENT NEURAL NETWORK

Neural language models often work best in combination with traditional $N$-gram models (Mikolov et al., 2011; Chelba et al., 2013; Williams et al., 2015; Ji et al., 2016; Shazeer et al., 2015), since the former excel at generalization while the latter ensure memorization. In addition, from initial experiments with memory-augmented neural language models, we found that usually only the previous five output representations are utilized. This is in line with observations by Tran et al. (2016). Hence, we experiment with a much simpler architecture depicted in Figure 1d. Instead of an attention mechanism, the output representations from the previous $N - 1$ time steps are directly used to calculate next-word probabilities. Specifically, at every time step we split the LSTM output into $N - 1$ vectors $[\boldsymbol{h}_t^1, \ldots, \boldsymbol{h}_t^{N-1}]$ and replace Equation 4 with

$$\boldsymbol{h}_t^* = \tanh\left(W^N \begin{bmatrix} \boldsymbol{h}_t^1 \\ \vdots \\ \boldsymbol{h}_{t-N+1}^{N-1} \end{bmatrix}\right) \qquad\qquad \in \mathbb{R}^k \qquad (13)$$

where $\boldsymbol{W}^N \in \mathbb{R}^{k \times (N-1)k}$ is a trainable projection matrix. This model is related to higher-order RNNs (Soltani & Jiang, 2016) with the difference that we do not incorporate output vectors from the previous steps into the hidden state but only use them for predicting the next word. Furthermore, note that at time step $t$ the first part of the output vector $\boldsymbol{h}_t^1$ will contribute to predicting the next word, the second part $\boldsymbol{h}_t^2$ will contribute to predicting the second word thereafter, and so on. As the output vectors from the $N - 1$ previous time-steps are used to score the next word, we call the resulting model an $N$-gram RNN.

## 3 RELATED WORK

Early attempts of using memory in neural networks have been undertaken by Taylor (1959) and Steinbuch & Piske (1963) by performing nearest-neighbor operations on input vectors and fitting parametric models to the retrieved sets. The dedicated use of external memory in neural architectures has more recently witnessed increased interest. Weston et al. (2015) introduced Memory Networks to explicitly segregate memory storage from the computation of the neural network, and Sukhbaatar et al. (2015) trained this model end-to-end with an attention-based memory addressing mechanism. The Neural Turing Machines by Graves et al. (2014) add an external differentiable memory with read-write functions to a controller recurrent neural network, and has shown promising results in simple sequence tasks such as copying and sorting. These models make use of external memory, whereas our model directly uses a short sequence from the history of tokens to dynamically populate an addressable memory.

In sequence modeling, RNNs such as LSTMs (Hochreiter & Schmidhuber, 1997) maintain an internal memory state as they process an input sequence. Attending over previous state outputs on top of an RNN encoder has improved performances in a wide range of tasks, including machine translation (Bahdanau et al., 2015), recognizing textual entailment (Rocktäschel et al., 2016), sentence summarization (Rush et al., 2015), image captioning (Xu et al., 2015) and speech recognition (Chorowski et al., 2015).

Recently, Cheng et al. (2016) proposed an architecture that modifies the standard LSTM by replacing the memory cell with a memory network (Weston et al., 2015). Another proposal for conditioning on previous output representations are Higher-order Recurrent Neural Networks (HORNNs, Soltani & Jiang, 2016). Soltani & Jiang found it useful to include information from multiple preceding RNN states when computing the next state. This previous work centers around preceding state vectors, whereas we investigate attention mechanisms on top of RNN *outputs*, *i.e.* the vectors used for predicting the next word. Furthermore, instead of pooling we use attention vectors to calculate a context representation of previous memories.

Yang et al. (2016) introduced a reference-aware neural language model where at every position a latent variable determines from which source a target token is generated, *e.g.*, by copying entries from a table or referencing entities that were mentioned earlier.

Another class of models that include memory into sequence modeling are Recurrent Memory Networks (RMNs) (Tran et al., 2016). Here, a memory block accesses the most recent input words to selectively attend over relevant word representations from a global vocabulary. RMNs use a global memory with two input word vector look-up tables for the attention mechanism, and consequently have a large number of trainable parameters. Instead, we proposed models that need much fewer parameters by *producing* the vectors that will be attended over in the future, which can be seen as a memory that is dynamically populated by the language model.

Finally, the functional separation of look-up keys and memory content has been found useful for Memory Networks (Miller et al., 2016), Neural Programmer-Interpreters (Reed & de Freitas, 2015), Dynamic Neural Turing Machines (Gulcehre et al., 2016), and Fast Associative Memory (Ba et al., 2016). We apply and extend this principle to neural language models.

## 4  EXPERIMENTS

We evaluate models on two different corpora for language modeling. The first is a subset of the Wikipedia corpus.[1] It consists of 7500 English Wikipedia articles (dump from 6 Feb 2015) belonging to one of the following categories: *People, Cities, Countries, Universities*, and *Novels*. We chose these categories as we expect articles in these categories to often contain references to previously mentioned entities. Subsequently, we split this corpus into a train, development, and test part, resulting in corpora of 22.5M words, 1.2M and 1.2M words, respectively. We map all numbers to a dedicated numerical symbol $N$ and restrict the vocabulary to the 77K most frequent words, encompassing 97% of the training vocabulary. All other words are replaced by the *UNK* symbol. The average length of sentences is 25 tokens. In addition to this Wikipedia corpus, we also run experiments on the Children's Book Test (CBT Hill et al., 2016). While this corpus is designed for cloze-style question-answering, in this paper we use it to test how well language models can exploit wider linguistic context.

### 4.1  TRAINING PROCEDURE

We use ADAM (Kingma & Ba, 2015) with an initial learning rate of 0.001 and a mini-batch size of 64 for optimization. Furthermore, we apply gradient clipping at a gradient norm of 5 (Pascanu et al., 2013). The bias of the LSTM's forget gate is initialized to 1 (Jozefowicz et al., 2016), while other parameters are initialized uniformly from the range $(-0.1, 0.1)$. Backpropagation Through Time (Rumelhart et al., 1985; Werbos, 1990) was used to train the network with 20 steps of unrolling. We reset the hidden states between articles for the Wikipedia corpus and between stories for CBT, respectively. We take the best configuration based on performance on the validation set and evaluate it on the test set.

## 5  RESULTS

In the first set of experiments we explore how well the proposed models and Tran et al.'s Recurrent-memory Model can make use of histories of varying lengths. Perplexity results for different attention window sizes on the Wikipedia corpus are summarized in Figure 2a. The average attention these models pay to specific positions in the history is illustrated in Figure 3. We observed that although our models attend over tokens further in the past more often than the Recurrent-memory Model, attending over a longer history does not significantly improve the perplexity of any attentive model.

The much simpler $N$-gram RNN model achieves comparable results (Figure 2b) and seems to work best with a history of the previous three output vectors (4-gram RNN). As a result, we choose the 4-gram model for the following $N$-gram RNN experiments.

---

[1]The wikipedia corpus is available at `https://goo.gl/s8cyYa`.

Figure 2: Perplexities of memory-augmented neural language models on the Wikipedia corpus (a-c) and accuracies on the CBT test set (d).

(a) Test perplexity of different attention architectures with varying attention window sizes. Best perplexity per model is italic.

| Model | Attention Window Size | | | |
|---|---|---|---|---|
| | 1 | 5 | 10 | 15 |
| RM(+tM-g) (Tran et al., 2016) | 83.5 | 80.5 | 80.3 | *80.1* |
| Attention | 82.2 | 82.2 | *82.0* | 82.8 |
| Key-Value | 78.7 | 79.0 | *78.2* | 78.9 |
| Key-Value-Predict | 76.1 | **75.8** | 76.0 | 75.8 |

(b) Comparison of $N$-gram neural language models. $w$ denotes the input size, $k$ the hidden size and $\theta_M$ the total number of model parameters.

| Model | $w$ | $k$ | $\theta_M$ | Dev | Test |
|---|---|---|---|---|---|
| 2-gram RNN | 300 | 564 | 23.9M | 76.0 | 77.1 |
| 3-gram RNN | 300 | 786 | 23.9M | 74.9 | 75.9 |
| 4-gram RNN | 300 | 968 | 23.9M | **74.8** | **75.9** |
| 5-gram RNN | 300 | 1120 | 23.9M | 76.0 | 77.3 |

(c) Summary of models with best attention window size $a$. The total number of model parameters, including word representations, is denoted by $\theta_{W+M}$ (without word representations $\theta_M$).

| Model | $w$ | $k$ | $a$ | $\theta_{W+M}$ | $\theta_M$ | Dev | Test |
|---|---|---|---|---|---|---|---|
| RNN | 300 | 307 | - | 47.0M | 23.9M | 121.7 | 125.7 |
| LSTM | 300 | 300 | - | 47.0M | 23.9M | 83.2 | 85.2 |
| FOFE HORNN (3-rd order) (Soltani & Jiang, 2016) | 300 | 303 | - | 47.0M | 23.9M | 116.7 | 120.5 |
| Gated HORNN (3-rd order) (Soltani & Jiang, 2016) | 300 | 297 | - | 47.0M | 23.9M | 93.9 | 97.1 |
| RM(+tM-g) (Tran et al., 2016) | 300 | 300 | 15 | 93.7M | 70.6M | 78.2 | 80.1 |
| Attention | 300 | 296 | 10 | 47.0M | 23.9M | 80.6 | 82.0 |
| Key-Value | 300 | 560 | 10 | 47.0M | 23.9M | 77.1 | 78.2 |
| Key-Value-Predict | 300 | 834 | 5 | 47.0M | 23.9M | **74.2** | **75.8** |
| 4-gram RNN | 300 | 968 | - | 47.0M | 23.9M | 74.8 | 75.9 |

(d) Results on CBT; those marked with ‡ are taken from Hill et al. (2016).

| Model | Named Entities | Common Nouns | Verbs | Prepositions |
|---|---|---|---|---|
| Humans (context+query) ‡ | 0.816 | 0.816 | 0.828 | 0.708 |
| Kneser-Ney LM ‡ | 0.390 | 0.544 | 0.778 | 0.768 |
| Kneser-Ney LM + cache ‡ | 0.439 | 0.577 | 0.772 | 0.679 |
| LSTM (context+query) ‡ | 0.418 | 0.560 | 0.818 | 0.791 |
| Memory Network ‡ | 0.666 | 0.630 | 0.690 | 0.703 |
| AS Reader, avg ensemble (Kadlec et al., 2016) | 0.706 | 0.689 | – | – |
| AS Reader, greedy ensemble (Kadlec et al., 2016) | 0.710 | 0.675 | – | – |
| QANN, 4 hops, GloVe (Weissenborn, 2016) | **0.729** | – | – | – |
| AoA Reader, single model (Cui et al., 2016a) | 0.720 | 0.694 | – | – |
| CAS Reader, mode avg (Cui et al., 2016b) | 0.692 | 0.657 | – | – |
| GA Reader, ensemble (Dhingra et al., 2016) | 0.719 | 0.694 | – | – |
| EpiReader, ensemble (Trischler et al., 2016) | 0.718 | **0.706** | – | – |
| FOFE HORNN (3-rd order) (Soltani & Jiang, 2016) | 0.465 | 0.497 | 0.774 | 0.741 |
| Gated HORNN (3-rd order) (Soltani & Jiang, 2016) | 0.508 | 0.547 | 0.790 | 0.774 |
| RM(+tM-g) (Tran et al., 2016) | 0.525 | 0.597 | 0.817 | 0.797 |
| LSTM | 0.523 | 0.604 | 0.819 | 0.786 |
| Attention | 0.538 | 0.595 | 0.826 | 0.803 |
| Key-Value | 0.528 | 0.601 | 0.822 | **0.813** |
| Key-Value-Predict | 0.528 | 0.599 | **0.829** | 0.803 |
| 4-gram RNN | 0.532 | 0.598 | 0.815 | 0.800 |

## 5.1 COMPARISON WITH STATE-OF-THE-ART MODELS

In the next set of experiments, we compared our proposed models against a variety of state-of-the-art models on the Wikipedia and CBT corpora. Results are shown in Figure 2c and 2d, respectively. Note that the models presented here do not achieve state-of-the-art on CBT as they are language models and not tailored towards cloze-sytle question answering. Thus, we merely use this corpus for comparing different neural language model architectures. We reimplemented the Recurrent-Memory model by Tran et al. (2016) with the temporal matrix and gating composition function (RM+tM-g).

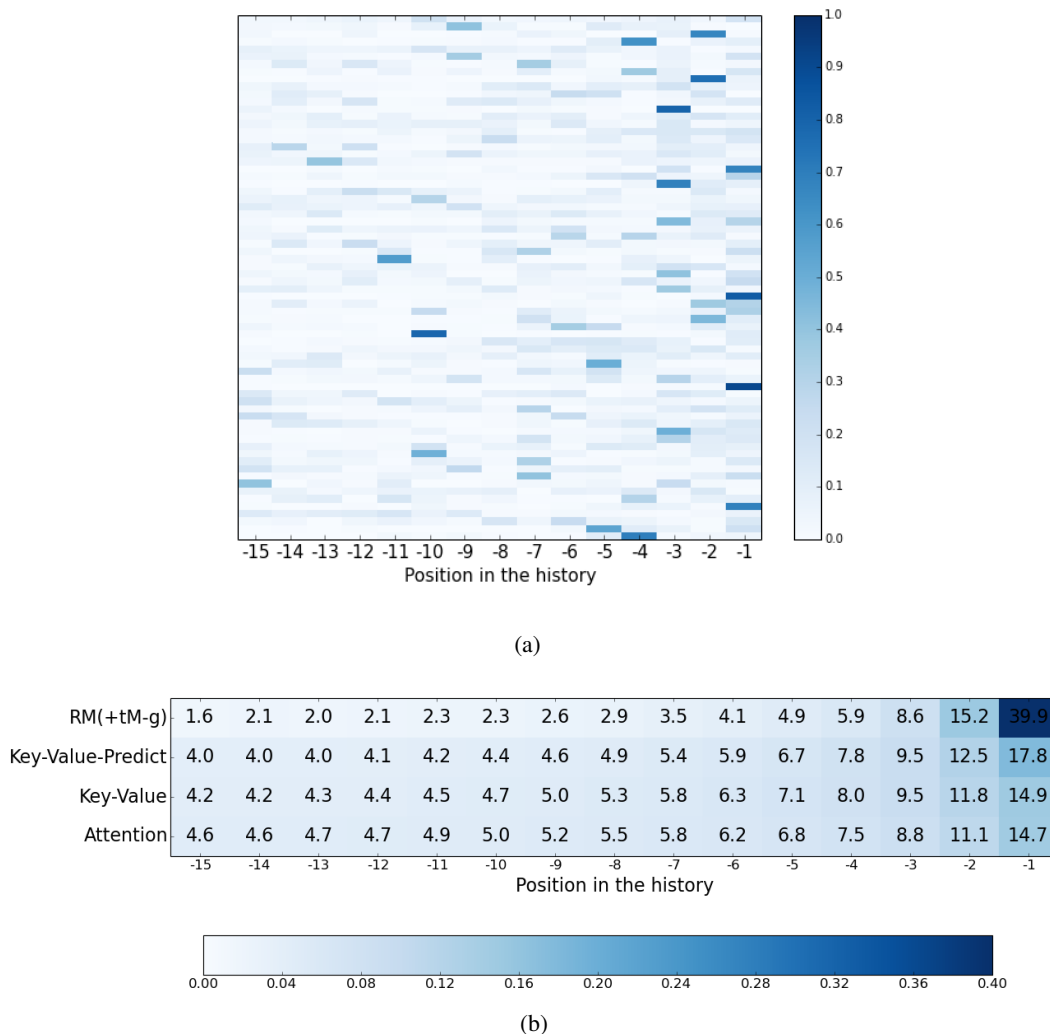

(a)

(b)

Figure 3: Attention weights of the Key-Value-Predict model on a randomly sampled Wikipedia article (a) and average attention weight distribution on the whole Wikipedia test set for RM(+tM-g), Attention, Key-Value and Key-Value-Predict models (b). The rightmost positions represent the most recent history.

Furthermore, we reimplemented Higher Order Recurrent Neural Networks (HORNNs) by Soltani & Jiang (2016).

To ensure a comparable number of parameters to a vanilla LSTM model, we adjusted the hidden size of all models to have roughly the same total number of model parameters. The attention window size $N$ for the $N$-gram RNN model was set to 4 according to the best validation set perplexity on the Wikipedia corpus. Below we discuss the results in detail.

**Attention** By using a neural language model with an attention mechanism over a dynamically populated memory, we observed a 3.2 points lower perplexity over a vanilla LSTM on Wikipedia, but only notable differences for predicting verbs and prepositions in CBT. This indicates that incorporating mechanisms for querying previous output vectors is useful for neural language modeling.

**Key-Value**    Decomposing the output vector into a key-value paired memory improves the perplexity by $7.0$ points compared to a baseline LSTM, and by $1.9$ points compared to the RM(+tM-g) model. Again, for CBT we see only small improvements.

**Key-Value-Predict**    By further separating the output vector into a key, value and next-word prediction part, we get the lowest perplexity and gain $9.4$ points over a baseline LSTM, a $4.3$ points compared to RM(+tM-g), and $2.4$ points compared to only splitting the output into a key and value. For CBT, we see an accuracy increase of $1.0$ percentage points for verbs, and $1.7$ for prepositions. As stated earlier, the performance of the Key-Value-Predict model does not improve significantly when increasing the attention window size. This leads to the conclusion that none of the attentive models investigated in this paper can utilize a large memory of previous token representations. Moreover, none of the presented methods differ significantly for predicting common nouns and named entities in CBT.

$N$**-gram RNN**    Our main finding is that the simple modification of using output vectors from the previous time steps for the next-word prediction leads to perplexities that are on par with or better than more complicated neural language models with attention. Specifically, the 4-gram RNN achieves only slightly worse perplexities than the Key-Value-Predict architecture.

## 6    CONCLUSION

In this paper, we observed that using an attention mechanism for neural language modeling where we separate output vectors into a key, value and predict part outperform simpler attention mechanisms on a Wikipedia corpus and the Children Book Test (CBT, Hill et al., 2016). However, we found that all attentive neural language models mainly utilize a memory of only the most recent history and fail to exploit long-range dependencies. In fact, a much simpler $N$-gram RNN model, which only uses a concatenation of output representations from the previous three time steps, is on par with more sophisticated memory-augmented neural language models. Training neural language models that take long-range dependencies into account seems notoriously hard and needs further investigation. Thus, for future work we want to investigate ways to encourage attending over a longer history, for instance by forcing the model to ignore the local context and only allow attention over output representations further behind the local history.

### ACKNOWLEDGMENTS

This work was supported by Microsoft Research and the Engineering and Physical Sciences Research Council through PhD Scholarship Programmes, an Allen Distinguished Investigator Award, and a Marie Curie Career Integration Award.

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
