# Peer review of "Frustratingly Short Attention Spans in Neural Language Modeling"

_ICLR 2017 — accepted_

[Official Review · AnonReviewer1 · rating 7 · confidence 4 · 15 Dec 2016]
**Review: Frustratingly Short Attention Spans in Neural Language Modeling**
originality 3 · substance 3

The paper presents an investigation of various neural language models designed to query context information from their recent history using an attention mechanism. The authors propose to separate the attended vectors into key, value and prediction parts. The results suggest that this helps performance. The authors also found that a simple model which which concatenates recent activation vectors performs at a similar level as the more complicated attention-based models.

The experimental methodology seems sound in general. I do have some issues with the way the dimensionality of the vectors involved in the attention-mechanism is chosen. While it’s good that the hidden layer sizes are adapted to ensure similar numbers of trainable parameters for all the models, this doesn’t control for the fact that key/value/prediction vectors of a higher dimensionality may simply work better regardless of whether their dimensions are dedicated to one particular task or used together. This separation clearly saves parameters but there could also be benefits of having some overlap of information assuming that vectors that lead to similar predictions may also be required in similar contexts for example. Some tasks may also require more dimensions than others and the explicit separation prevents the model from discovering and exploiting this. 

While memory augmented RNNs and RNNs with attention mechanisms are not new, some of these architectures had not yet been applied to language modeling. Similarly (and as acknowledged by the authors), the strategy of separating key and value functionality has been proposed before, but not in the context of natural language modeling. I’m not sure about the novelty of the proposed n-gram RNN because I recall seeing similar architectures before but I understand that novelty was not the point of that architecture as it mainly serves as a proof of the lack of ability of the more complicated architectures to do better. In that sense I do consider it an inventive baseline that could be used in future work to test the ability of other models that claim to exploit long-term dependencies. 

The exact computation of the representation h_t was initially not that clear to me (the terms hidden and output can be ambiguous at times) but besides this, the paper is quite clear and generally well-written.

The results in this paper are important because they show that learning long-term dependencies is not a solved problem by any means. The authors provide a very nice comparison to prior results and the fact that their n-gram RNN is often at least competitive with far more complicated approaches is a clear indication that some of those methods may not capture as much context information as previously thought. The success of the separation of key/value/prediction functionality in attention-based system is also noteworthy although I think this is something that needs to be investigated more thoroughly (i.e., with more control for hyperparameter choices). 


Pros:
Impressive and also interesting results.
Good comparison with earlier work.
The n-gram RNN is an interesting baseline.


Cons:
The relation between the attention-mechanism type and the number of hidden units weakens the claim that the key/value/prediction separation is the reason for the increase in performance somewhat.
The model descriptions are not entirely clear.
I would have liked to have seen what happens when the attention is applied to a much larger context size.

[Official Review · AnonReviewer3 · rating 7 · confidence 4 · 17 Dec 2016]
clarity 2 · meaningful comparison 1 · recommendation (unofficial) 1

This paper explores a variety of memory augmented architectures (key, key-value, key-predict-value) and additionally simpler near memory-less RNN architectures. Using an attention model that has access to the various decompositions is an interesting idea and one worth future explorations, potentially in different tasks where this type of model could excel even more. The results over the Wikipedia corpus are interesting and feature a wide variety of different model types. This is where the models suggested in the paper are strongest. The same models run over the CBT dataset show a comparable but less convincing demonstration of the variations between the models.

The authors also released their Wikipedia corpus already. Having inspected it I consider it a positive and interesting contribution. I still believe that, if a model was found that could better handle longer term dependencies, it would do better on this Wikipedia dataset, but at least within the realm of what . As an example, the first article in train.txt is about a person named "George Abbot", yet "Abbot" isn't mentioned again until the next sentence 40 tokens later, and then the next "Abbot" is 15 tokens from there. Most gaps between occurrences of "Abbot" are dozens of timesteps. Performing an analysis based upon easily accessed information, such as when the same token reappears again or average sentence length, may be useful as an approximation for the length that an attention window may prefer.

This is a well explained paper that raises interesting questions regarding the spans used in existing language modeling approaches and serves as a potential springboard for future directions.

[Official Review · AnonReviewer2 · rating 7 · confidence 4 · 19 Dec 2016]
soundness 5 · originality 5 · clarity 3 · impact 4 · substance 4 · recommendation (unofficial) 3

This paper focusses on attention for neural language modeling and has two major contributions:

1. Authors propose to use separate key, value, and predict vectors for attention mechanism instead of a single vector doing all the 3 functions. This is an interesting extension to standard attention mechanism which can be used in other applications as well.
2. Authors report that very short attention span is sufficient for language models (which is not very surprising) and propose an n-gram RNN which exploits this fact.

The paper has novel models for neural language modeling and some interesting messages. Authors have done a thorough experimental analysis of the proposed ideas on a language modeling task and CBT task.

I am convinced with authors’ responses for my pre-review questions.

Minor comment: Ba et al., Reed & de Freitas, and Gulcehre et al. should be added to the related work section as well.

[Public Comment · Xiang Zhang · 17 Feb 2017]
**Nice read!**

(I read the paper after knowing it got accepted to ICLR 2017)

This paper is quite illuminating to me, as it shows that language modeling may not be the right task to show whether a model has the ability to hold and use long-term information. However, it is up for debate whether that is because of the specific datasets used in the paper.

[Final Decision · Program Chairs · 06 Feb 2017]
**ICLR committee final decision**

Reviewers found this paper to be a rigorous and "thorough experimental analysis" of context-length in language modelingv through the lens of an "interesting extension to standard attention mechanism". The paper reopens and makes more problematic widely accepted but rarely verified claims of the importance of long-term dependency.
 
 Pros:
 - "Well-explained" and clear presentation
 - Use of an "inventive baseline" in the form a ngram rnn 
 - Use a impactful corpus for long-term language modeling
 
 Cons: 
 - Several of the ideas have been explored previously.
 - Some open questions about the soundness of parameters (rev 1)
 - Requests for deeper analysis on data sets released with the paper.